# AugKD: Ingenious Augmentations Empower Knowledge Distillation for Image Super-Resolution

**Yun Zhang**[*]
The Hong Kong University of Science and Technology

**Wei Li**[*], **Simiao Li**, **Hanting Chen**, **Zhijun Tu**
Huawei Noah's Ark Lab

**Bingyi Jing**
Southern University of Science and Technology

**Shaohui Lin**
East China Normal University

**Jie Hu**[†]
Huawei Noah's Ark Lab

**Wenjia Wang**[†]
The Hong Kong University of Science and Technology (Guangzhou)

## Abstract

Knowledge distillation (KD) compresses deep neural networks by transferring task-related knowledge from cumbersome pre-trained teacher models to more compact student models. However, vanilla KD for image super-resolution (SR) networks yields only limited improvements due to the inherent nature of SR tasks, where the outputs of teacher models are noisy approximations of high-quality label images. In this work, we show that the potential of vanilla KD has been underestimated and demonstrate that the ingenious application of data augmentation methods can close the gap between it and more complex, well-designed methods. Unlike conventional training processes typically applying image augmentations simultaneously to both low-quality inputs and high-quality labels, we propose AugKD utilizing unpaired data augmentations to 1) generate auxiliary distillation samples and 2) impose label consistency regularization. Comprehensive experiments show that the AugKD significantly outperforms existing state-of-the-art KD methods across a range of SR tasks.

## 1 Introduction

SR is an essential yet challenging task in computer vision (CV) that focuses on reconstructing high-resolution (HR) image from its low-resolution (LR) counterpart (Lim et al., 2017; Zhang et al., 2018). In recent years, the convolutional neural networks (CNNs) and Transformers (Liang et al., 2021; Yang et al., 2020; Wang et al., 2022b; Zamir et al., 2022) have achieved significant success in SR. However, despite the impressive performance of deep learning-based SR models, their practical deployment is often constrained by the high computational and memory requirements (Zhang et al., 2021c). As a result, there has been an increasing focus on developing SR model compression techniques to enable their use in real-world applications, especially for resource-constrained devices.

KD is an effective technique for reducing computational costs and memory requirements during model deployment, while also enhancing the performance of student models. It works by transferring the "dark knowledge" from a well-performing but computationally heavy teacher model to a more lightweight student model (Gao et al., 2019; Hui et al., 2019; Zhang et al., 2021b). Compared to other model compression techniques, such as quantization (Gupta et al., 2015; Hubara et al., 2016; Ignatov et al., 2021; Wu et al., 2016), pruning (Anwar et al., 2017; Wang et al., 2021b; Liu et al., 2019), and neural architecture search (NAS) (Wu et al., 2019; Howard et al., 2019; Guo et al., 2020), KD has garnered significant attention due to its outstanding performance and wide applicability.

---

[*]Co-first author.
[†]Corresponding author. hujie23@huawei.com, wenjiawang@hkust-gz.edu.cn

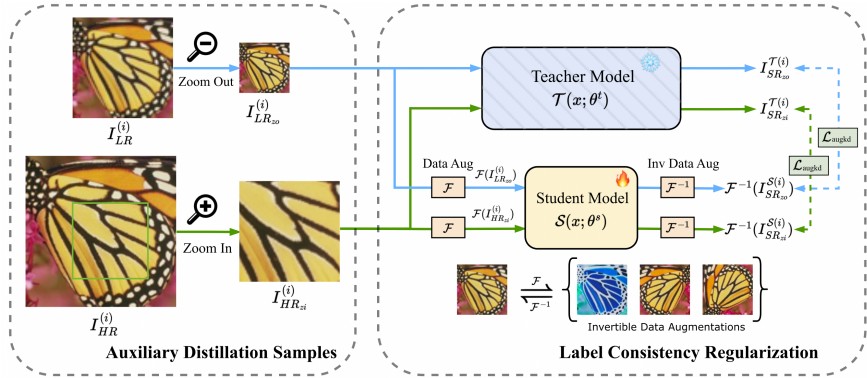

Figure 1: Framework of the AugKD method. Facilitates the transfer of knowledge through the auxiliary distillation samples and label consistency regularization.

The effectiveness of KD has been well-established in natural language processing (NLP) (Gou et al., 2021; Sanh et al., 2019) and conventional high-level CV tasks, such as classification, detection, and segmentation (Hinton et al., 2015; Park et al., 2019; Tung & Mori, 2019). However, its application to SR tasks remains relatively underexplored (He et al., 2020; Wang et al., 2021c; Zhang et al., 2021b; Lee et al., 2020). The use of standard response-based KD methods (Hinton et al., 2015) or those optimized for high-level CV tasks (Romero et al., 2014; Yim et al., 2017; Zagoruyko & Komodakis, 2016) typically results in only marginal improvements and may even have negative effects when applied to distilling SR networks, as observed by He et al. (2020) and confirmed by our experiments in section 4. Previous KD methods specifically designed for SR are mostly feature-based, where the student model is forced to mimic the intermediate features of the teacher model directly (He et al., 2020) or through a pre-trained perceptual feature extractor like VGG (Yao et al., 2022a; Wang et al., 2021c). However, these feature-based approaches have limited applicability. In practice, the architecture of teacher models is often inaccessible due to commercial, privacy, or safety restrictions, making feature-based methods impractical for some real-world applications.

For the knowledge distillation of super-resolution models, most of previous explanations for the mechanism of KD no longer hold due to the unique task characteristics. Since the teacher model's output, as a noisy approximation to the ground-truth (GT) high-quality image, contains barely extra information exceeding GT, the "dark knowledge" of teacher are hardly transferred to student model through KD. Distinct to exist feature-based methods, we propose AugKD, an alternative approach to enhance the knowledge distillation via the data augmentations. It shifts the paradigm from developing various knowledge types (Gou et al., 2021) to more task-adapted training data mining and construction with effective utilization of pre-trained teacher. Specifically, the AugKD consists of two major modules: *auxiliary distillation sample generation* and *label consistency regularization*. The auxiliary training examples are built from (LR, HR) pairs by zoom-in and zoom-out augmentations. Then teacher model is able to guide the student model with these image samples. It frees the teacher model from merely echoing the GT labels inaccurately. Moreover, we realize the label consistency regularization into the KD for SR by defining several *invertible data augmentation* operations. The student model is forced to yield the same output as the teacher model, given the augmented inputs. The regularization makes the student model exposed to a diverse range of inputs, substantially improving performance (Oliver et al., 2018; Jeong et al., 2019; Englesson & Azizpour, 2021). The AugKD is logits-based and independent of network architectures. It shows great universality among a diverse array of SR model families and SR tasks. In summary, our main contributions are three-fold:

- We analyze the mechanisms of KD for SR, and propose AugKD adapted to the unique task properties. It facilitates the student model's learning by auxiliary training samples.

- We leverage the label consistency regularization into KD for SR by specifying several invertible data augmentations. It improves the model's generalizability.

- The proposed AugKD, applies broadly to multiple teacher-student configurations, promising a cutting-edge KD approach for SR.

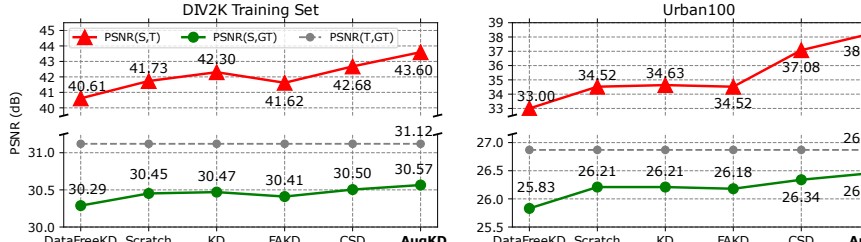

Figure 2: Similarity between the student and teacher ×4 EDSR models under different training approaches (x-axis). PSNR(S,T) represents the average PSNR between the student and teacher outputs, with higher values indicating greater similarity. PSNR(S,GT) shows the average PSNR between the student output and ground-truth HR image, with higher values indicating better fitting (left: training set) or generalization (right: testing set).

## 2 RELATED WORKS

### 2.1 IMAGE SUPER-RESOLUTION

Deep neural networks (DNNs) have achieved impressive success in image SR. Dong et al. (2014) introduced a CNN SR model with only three convolutional layers first. Subsequently, residual learning was incorporated into the VDSR model (Kim et al., 2016), which expanded the network to 20 convolutional layers. Lim et al. (2017) proposed EDSR, which utilized simplified residual blocks (He et al., 2016), and Zhang et al. (2018) introduced the even deeper RCAN network. These methods, among others, have set state-of-the-art performance benchmarks by increasing network depth and width. More recently, Transformers have gained significant attention in the field of image restoration. The SwinIR model (Liang et al., 2021) applies the Swin Transformer architecture for deep feature extraction. The Restormer model (Zamir et al., 2022) proposed a hierarchical multi-scale structure, introducing a more efficient Transformer blocks that alters the attention mechanisms and the feed-forward network. While CNNs and Transformers models demonstrate extraordinary effect in SR, they are often associated with high memory and computational costs.

### 2.2 KNOWLEDGE DISTILLATION

**KD for high-level CV**. Knowledge distillation is widely recognized as an effective model compression method that can significantly reduce the computation overload and improve student's capability (Hinton et al., 2015; Yim et al., 2017; Gou et al., 2021). The response-based KD methods are simple yet effective where the student models directly imitate the predictions or logits of the teacher model (Hinton et al., 2015; Zhao et al., 2022; Chen et al., 2017). The proposed AugKD method falls into this category since only the final outputs of models are aligned. Besides the output of the networks, the intermediate features can also be used to improve the student model, by matching feature maps directly, after dimension standardization (Zagoruyko & Komodakis, 2016) or extra modules (Kim et al., 2018; Passban et al., 2021; Guo et al., 2021). The relations between layers or samples can be used for KD, such as correlation (Yim et al., 2017; You et al., 2017), mutual information (Passalis et al., 2020), or pairwise or triple-wise geometric relations (Park et al., 2019).

**KD for super resolution**. Lately, there has been an increasing number of efforts made on the KD for super-resolution networks. He et al. (2020) proposed the FAKD to align the dimensions of models' feature maps by spatial affinity matrix to train the student model. Lee et al. (2020) employed an encoder to extract the compact features from HR images to initialize the generator network and thereby perform feature distillation. Wang et al. (2021c) proposed a channel-sharing self-distillation method with perceptual contrastive losses. To train SR network under the privacy and data transmission limitations, Zhang et al. (2021b) employed a generator to support data-free KD. MTKDSR (Yao et al., 2022b) employed two teacher models with different SR objectives (PSNR, perceptual) to guide the student model simultaneously. CrossKD (Fang et al., 2023) divides the teacher and student networks into two segments that are interchanged and connected to perform

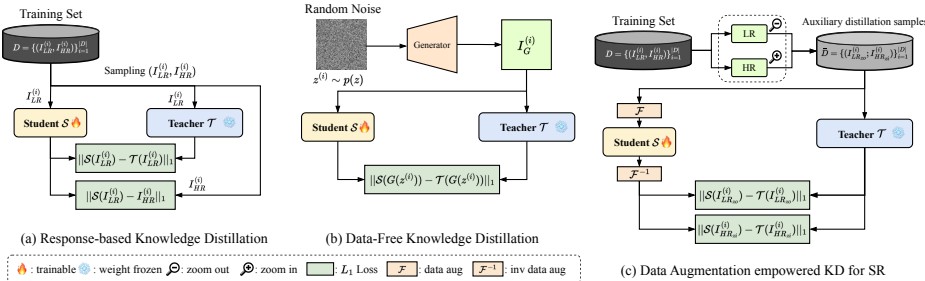

Figure 3: Comparison between the logits-KD, Data Free KD, and AugKD. The first two fail to enable the function of teacher model.

forward propagation. The common limitation of these methods is that they are only applicable to CNN-based models and have certain requirements on the teacher-student structure.

## 3 METHODOLOGY

### 3.1 NOTATIONS AND PRELIMINARIES

Let $\mathcal{T}(x;\theta^t)$ and $\mathcal{S}(x;\theta^s)$ be a teacher and a student SR model with parameters $\theta^t$ and $\theta^s$ for the super-resolution of input $x$, respectively. Given an input LR image $I_{LR}^{(i)} \in \mathbb{R}^{H \times W \times 3}$, the output SR images of the two networks are denoted by $I_{SR}^{\mathcal{T}(i)} = \mathcal{T}(I_{LR}^{(i)};\theta^t) \in \mathbb{R}^{s_c H \times s_c W \times 3}$ and $I_{SR}^{\mathcal{S}(i)} = \mathcal{S}(I_{LR}^{(i)};\theta^s) \in \mathbb{R}^{s_c H \times s_c W \times 3}$, where $H \times W$ is the input size and $s_c \in \mathbb{Z}^+$ is the scaling factor. The L1-norm reconstruction loss is computed as:

$$\mathcal{L}_{rec} = \|I_{SR}^{\mathcal{S}(i)} - I_{HR}^{(i)}\|_1, \tag{1}$$

where $I_{HR}^{(i)}$ is the ground-truth HR label. And the vanilla response-based KD loss is given by

$$\mathcal{L}_{kd} = \|I_{SR}^{\mathcal{S}(i)} - I_{SR}^{\mathcal{T}(i)}\|_1, \tag{2}$$

which is computed directly by the output of teacher and student models.

### 3.2 MOTIVATION

Since the introduction of knowledge distillation by Hinton et al. (2015), numerous studies have analyzed and discussed the mechanisms through which teacher supervision enhances the performance of student models (Tang et al., 2020; Stanton et al., 2021; Wang et al., 2021a; Zhang et al., 2022; Harutyunyan et al., 2023). It is widely accepted that in response-based KD, the "dark knowledge" from the teacher model encompasses the inter-class and inter-example relational information found in the output logits, which is not present in the ground-truth labels.

However, there is barely such a benefit in SR tasks that reconstruct image pixels. Since the outputs of the SR network $I_{SR}^{T(i)}$ are noisy and inaccurate approximations of the ground-truth distribution of high-resolution image $I_{HR}^{(i)}$, as shown in Figure 3 (a). Directly aligning the model outputs hardly transfers knowledge and may even mislead the student model. The guide capability of the teacher model is shaded by $I_{HR}^{(i)}$, resulting in limited KD effects. To verify this hypothesis, we train an EDSR network of scale ×4 using different training methods (data-free KD, supervised training without distillation, Logits-KD, FAKD, CSD and the proposed AugKD). To make the models comparable, the data-free KD uses the LR of the training set and discards HR, assuming that there is an oracle image generator $G$. Then we compute the PSNR metrics between the outputs of teacher and student models, on the training and testing sets, respectively. It reflects the similarities between networks, and the extent to which the student model is impacted by the teacher model. The results shown in Figure 2 indicate that the existing KD approaches make the student model performs more like teacher only to a small extent, since the the improvements of PSNR(S,T) over training without KD are limited. Therefore, the PSNR referring to GT, PSNR(S,GT), are also low on both training and testing sets.

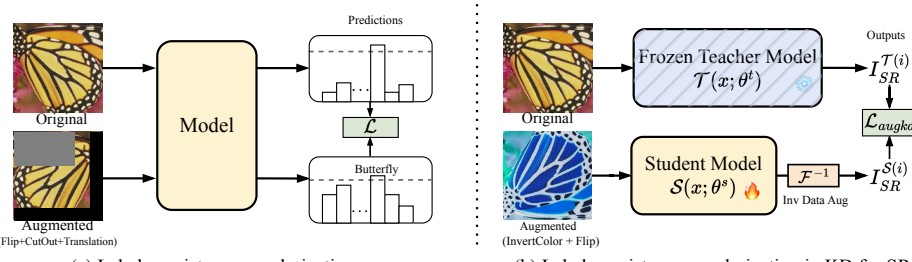

(a) Label consistency regularization  (b) Label consistency regularization in KD for SR

Figure 4: Comparison of the label consistency regularization in high-level CV and KD for SR. The augmentations should be invertible to make the models' output comparable.

This issue cannot be addressed by simple data augmentation that reuses the training image pairs to produce augmented LR and HR with pairwise rotations or flips, i.e. $(I_{LR}, I_{HR}) \Rightarrow (I_{LR}^{aug}, I_{HR}^{aug})$. The "recycled" data are inadequate to enable the function of teacher model. The data-free knowledge distillation methods (Zhang et al., 2021b) stay out of this problem due to the discard of HR references from the training data. The supervision signals solely come from the teacher model, as illustrated in Figure 3 (b). Although teacher model's knowledge are transferred to student model (it's only supervised by the teacher model's output), it's impractical to discard the labels from training data especially when they are available. Besides, the teacher model may yield noisier output on the generated training images.

Above findings motivate us to build a more task-adapted KD framework by mining information from the training data. Specifically, we construct auxiliary training inputs through data augmentations to function the KD. These data are closely related with the training set to prevent distributional bias. And we introduce the label consistency regularization through invertible data augmentations.

### 3.3 AUXILIARY DISTILLATION SAMPLES

Since the teacher model's knowledge is shaded by the ground-truth HR labels, we perform knowledge distillation by extra LR images rather than the raw training data. To make the generation process efficient and the generated images distributed closely with the original, the auxiliary training samples are obtained from original LR, HR pairs, as demonstrated in Figure 1. Two image zooming operations are employed: The zoom-in ⊕ operation is facilitated by randomly cropping patches from $I_{HR}^{(i)}$. The cropped patches have the same size as the LR image $I_{LR}^{(i)}$, for the convenience of batch processing. Conversely, the zoom-out ⊖ operation is carried out by down-sampling the LR image in the same manner of degradation as $I_{LR}^{(i)}$. The two obtained auxiliary LR images are denoted as $I_{LR_{zo}}^{(i)} \in \mathbb{R}^{H/s_c \times W/s_c \times 3}$ and $I_{LR_{zi}}^{(i)} \in \mathbb{R}^{H \times W \times 3}$.

For a pair of original training examples $(I_{LR}^{(i)}, I_{HR}^{(i)})$, the output of the zoom-out operation is unique, but the zoom-in operation on $I_{HR}^{(i)}$ could result in various outcomes according to the strategy of patch selection. Beyond random cropping, regions can also be selected based on their reconstruction difficulty or texture complexity. It's observed in our experiments that adapted selection would incur a higher computational cost with marginal performance gains.

After generating auxiliary distillation samples, the teacher model provides corresponding SR images for the zoom-in and zoom-out LR inputs to supervise the student model. Since there is only the teacher model's supervision for these training samples, teacher's distribution information are unshaded from GT and able to impact the student model. AugKD provides a more refined and data-centric approach to KD, reflecting its benefits in effective data utilization and superior performance in SR tasks. The overall loss is constructed by adding an extra loss term that is computed on the auxiliary distillation samples to the reconstruction loss (Equation (1)) and conventional KD loss (Equation (2)),

$$\mathcal{L}_{augkd} = \|I_{SR_{zo}}^{\mathcal{S}(i)} - I_{SR_{zo}}^{\mathcal{T}(i)}\|_1 + \|I_{SR_{zi}}^{\mathcal{S}(i)} - I_{SR_{zi}}^{\mathcal{T}(i)}\|_1, \tag{3}$$

where $I_{SR_{zo}}^{\mathcal{S}(i)} = \mathcal{S}(I_{LR_{zo}}^{(i)}; \theta^s)$, $I_{SR_{zo}}^{\mathcal{T}(i)} = \mathcal{T}(I_{LR_{zo}}^{(i)}; \theta^t)$ and the other terms are computed similarly. If zoom-out is performed, we compute the reconstruction loss between $I_{SR_{zo}}^{\mathcal{S}(i)}$ and $I_{LR}^{(i)}$ also. To sum up,

$$\mathcal{L} = \mathcal{L}_{rec} + \lambda_{kd}\mathcal{L}_{kd} + \lambda_{augkd}\mathcal{L}_{augkd}, \tag{4}$$

Table 1: SR model specifications and statistics (×4 scale). The FLOPs and frames per second (FPS) are computed with a 3×256×256 input image on single V100 GPU of 64GB VRAM. The block denotes the number of residual blocks for EDSR and RCAN (in each residual group) or Swin transformer blocks for SwinIR models.

| Model | Role | Network | | | FLOPs (G) | #Params | FPS |
|---|---|---|---|---|---|---|---|
| | | Channel | Block | Group | | | |
| EDSR | Teacher | 256 | 32 | - | 3293.35 | 43.09 M | 3.233 |
| | Student | 64 | 32 | - | 207.28 | 2.70 M | 33.958 |
| RCAN | Teacher | 64 | 20 | 10 | 1044.03 | 15.59 M | 6.162 |
| | Student | 64 | 6 | 10 | 366.98 | 5.17 M | 12.337 |
| SwinIR | Teacher | 180 | 6 | - | 861.27 | 11.90 M | 0.459 |
| | Student | 60 | 4 | - | 121.48 | 1.24 M | 0.874 |

where $\lambda_{kd}$ and $\lambda_{augkd}$ are the loss weights.

## 3.4 LABEL CONSISTENCY REGULARIZATION

Consistency regularization is commonly used in semi-supervised and self-supervised learning. As illustrated in Figure 4 (a), it encourages the prediction of the network to be consistent over perturbed training examples, leading to robustness against corrupted data in test time (Oliver et al., 2018; Englesson & Azizpour, 2021; Jeong et al., 2019). The model is trained to identify the crucial semantic information related to specific tasks from the input images, despite the possible noise and perturbations. The regularization is based on various image augmentation techniques, like rotation, shearing, cutout, and translation.

KD encourages the student model to produce the same predictions as the teacher model. Such characteristic should hold even their inputs are differently augmented, as the task-related semantic information remains unchanged and the input perturbations should not significantly distinguish between the outputs of the teacher and student models. To realize label consistency regularization, we apply data augmentations on the input of student model while keeping the teacher model's input unperturbed. This approach allows the student to learn invariant representations across diverse transformations. Meanwhile, the student is guided by a more powerful teacher model, whose supervision are from non-perturbed inputs that inherently provide superior quality compared to those from augmented inputs. Thereby the auxiliary distillation samples and the teacher model are fully leveraged. Taking the zoom-in LR sample as an example, the consistency regularization can be represented as:

$$\mathcal{L} = ||\mathcal{S}(\mathcal{F}(I_{HR_{zi}}); \theta^s) - \mathcal{T}(I_{HR_{zi}}; \theta^t)||_1,$$

where $\mathcal{F}(\cdot)$ denotes the perturbation function.

However, as super resolution is a pixel-level image-to-image CV task that is weakly relevant to semantic information of image subject, any tweak on the input can alter the model's output. For KD, the student model's output would consequently be incomparable with the teacher model's. Therefore, we need to perform inverse augmentation, namely $\mathcal{F}^{-1}(\cdot)$, on the output of the student model. The label consistency regularization becomes:

$$\mathcal{L} = ||\mathcal{F}^{-1}(\mathcal{S}(\mathcal{F}(I_{HR_{zi}}); \theta^s)) - \mathcal{T}(I_{HR_{zi}}; \theta^t)||_1.$$

The selected augmentations should be invertible and relevant to the SR task for maintaining the crucial pixel-level details after augmentation. It requires that for any input image $I$, $\mathcal{F}^{-1}(\mathcal{F}(I)) = I$. Hence, a number of popular image augmentations, such as blurring, cutout, brightness adjustment, and cropping, are not applicable as they do not meet this prerequisite. Instead, we employ two geometric transformations, horizontal/vertical flip and 90°/180°/270° rotations, along with the *color inversion* transformation that subtracts each pixel intensity value of the input image from 255 (or 1 if normalized): $\mathcal{F}(I) = 255 - I$. The color inversion is invertible and maintains the relative magnitude among pixel values. It also prompts the student models to be more sensitive to essential structural features such as lines and edges. Right bottom of Figure 1 illustrates the three types of invertible data augmentations employed to realize label consistency.

Table 2: Quantitative comparison (average PSNR/SSIM) between AugKD and other distillation methods for **EDSR** of three SR scales. The best and second-best performances are highlighted in bold and underlined, respectively. An asterisk indicates that the results in a row are from reproduction.

| Scale | Method | Set5 | Set14 | BSD100 | Urban100 |
|-------|--------|------|-------|--------|----------|
| | | PSNR/SSIM | PSNR/SSIM | PSNR/SSIM | PSNR/SSIM |
| ×2 | Scratch | 38.00/0.9605 | 33.57/0.9171 | 32.17/0.8996 | 31.96/0.9268 |
| | KD | 38.04/0.9606 | 33.58/0.9172 | 32.19/0.8998 | 31.98/0.9269 |
| | FitNet | 37.59/0.9589 | 33.09/0.9136 | 31.79/0.8953 | 30.46/0.9111 |
| | AT | 37.96/0.9603 | 33.48/0.9167 | 32.12/0.8990 | 31.71/0.9241 |
| | RKD | 38.03/0.9606 | 33.57/0.9173 | 32.18/0.8998 | 31.96/0.9270 |
| | FAKD* | 37.99/0.9606 | 33.60/0.9173 | 32.19/0.8998 | 32.04/0.9275 |
| | CSD* | 38.06/0.9607 | 33.65/0.9179 | 32.22/0.9004 | 32.26/0.9300 |
| | **AugKD** | **38.15/0.9610** | **33.80/0.9195** | **32.27/0.9007** | **32.53/0.9320** |
| ×3 | Scratch | 34.39/0.9270 | 30.32/0.8417 | 29.08/0.8046 | 27.99/0.8489 |
| | KD | 34.43/0.9273 | 30.34/0.8422 | 29.10/0.8050 | 28.00/0.8491 |
| | FitNet | 33.35/0.9178 | 29.71/0.8323 | 28.62/0.7949 | 26.61/0.8167 |
| | AT | 34.29/0.9262 | 30.26/0.8406 | 29.03/0.8035 | 27.76/0.8443 |
| | RKD | 34.43/0.9274 | 30.33/0.8423 | 29.09/0.8051 | 27.96/0.8493 |
| | FAKD* | 34.39/0.9272 | 30.34/0.8426 | 29.10/0.8052 | 28.07/0.8511 |
| | CSD* | 34.45/0.9275 | 30.32/0.8430 | 29.11/0.8061 | 28.21/0.8549 |
| | **AugKD** | **34.59/0.9287** | **30.47/0.8448** | **29.20/0.8073** | **28.44/0.8578** |
| ×4 | Scratch | 32.29/0.8965 | 28.68/0.7840 | 27.64/0.7380 | 26.21/0.7893 |
| | KD | 32.30/0.8965 | 28.70/0.7842 | 27.64/0.7382 | 26.21/0.7897 |
| | FitNet | 31.65/0.8873 | 28.33/0.7768 | 27.38/0.7309 | 25.40/0.7637 |
| | AT | 32.22/0.8952 | 28.63/0.7825 | 27.59/0.7365 | 25.97/0.7825 |
| | RKD | 32.30/0.8965 | 28.69/0.7842 | 27.64/0.7383 | 26.20/0.7899 |
| | FAKD* | 32.27/0.8960 | 28.65/0.7836 | 27.62/0.7379 | 26.18/0.7895 |
| | CSD | 32.34/0.8974 | 28.72/0.7856 | 27.68/0.7396 | 26.34/0.7948 |
| | **AugKD** | **32.47/0.8981** | **28.80/0.7866** | **27.71/0.7403** | **26.45/0.7963** |

## 4 EXPERIMENTS

### 4.1 EXPERIMENT SETUPS

**Backbones and Baselines.** We use EDSR (Lim et al., 2017), RCAN (Zhang et al., 2018), and SwinIR (Liang et al., 2021) as backbone models to evaluate AugKD and compare it with existing KD methods. The specifications of the teacher and student networks, along with statistics such as FLOPs, number of parameters, and inference speed (FPS), are shown in Table 1. We compare AugKD with the following baseline methods: training from scratch, response-based KD (Hinton et al., 2015), FitNet (Romero et al., 2014), AT (Zagoruyko & Komodakis, 2016), RKD (Park et al., 2019), FAKD (He et al., 2020), CSD (Wang et al., 2021c), and CrossKD (Fang et al., 2023). While FitNet, AT, and RKD were originally designed for high-level CV tasks, they are also applicable to SR models. However, CSD, being a self-distillation method, is not suitable for distilling RCAN (depth compression) and SwinIR (transformer-based) models. Performance is evaluated using peak signal-to-noise ratio (PSNR) and structural similarity index (SSIM) on the Y channel of the YCbCr color space. Detailed training settings are provided in Appendix A.1.

### 4.2 RESULTS AND COMPARISON

**Comparison with Baseline Methods.** The quantitative results (PSNR / SSIM) for training EDSR, RCAN, and SwinIR networks are presented in Table 2, 3, and 10 respectively, for ×2, ×3, and ×4 scales. The following conclusions can be drawn from these results: (**1**) Existing KD methods have limited benefits and some even result in student models worse than those trained without KD. For instance, EDSR models trained with FAKD sometimes underperform the ones trained from scratch. It showcases that the "dark knowledge" cannot be directly simply transferred to the student SR model. (**2**) The KD methods initially designed for high-level CV tasks (FitNet, AT, RKD), while applicable, hardly improve the SR models over training from scratch. It's caused by the intrinsic difference

Table 3: Quantitative comparison between AugKD and other distillation methods for **RCAN**. The best and second-best performances are highlighted in bold and underlined, respectively.

| Scale | Method | Set5 PSNR/SSIM | Set14 PSNR/SSIM | BSD100 PSNR/SSIM | Urban100 PSNR/SSIM |
|---|---|---|---|---|---|
| ×2 | Scratch | 38.13/0.9610 | 33.78/0.9194 | 32.26/0.9007 | 32.63/0.9327 |
| | KD | 38.18/0.9611 | 33.83/0.9197 | 32.29/0.9010 | 32.67/0.9329 |
| | FitNet | 37.97/0.9602 | 33.57/0.9174 | 32.19/0.8999 | 32.06/0.9279 |
| | AT | 38.13/0.9610 | 33.70/0.9187 | 32.25/0.9005 | 32.48/0.9313 |
| | RKD | 38.18/0.9612 | 33.78/0.9191 | 32.29/0.9011 | 32.70/0.9330 |
| | FAKD* | 38.17/0.9612 | 33.83/0.9199 | 32.29/0.9011 | 32.65/0.9330 |
| | CrossKD | 38.18/0.9612 | 33.81/0.9194 | 32.30/0.9011 | 32.66/0.9332 |
| | **AugKD** | **38.23/0.9614** | **33.90/0.9201** | **32.33/0.9016** | **32.87/0.9349** |
| ×3 | Scratch | 34.61/0.9288 | 30.45/0.8444 | 29.18/0.8074 | 28.59/0.8610 |
| | KD | 34.61/0.9291 | 30.47/0.8447 | 29.21/0.8080 | 28.62/0.8612 |
| | FitNet | 34.21/0.9248 | 30.20/0.8399 | 29.05/0.8044 | 27.89/0.8472 |
| | AT | 34.55/0.9287 | 30.43/0.8438 | 29.17/0.8070 | 28.43/0.8577 |
| | RKD | 34.67/0.9292 | 30.48/0.8451 | 29.21/0.8080 | 28.60/0.8610 |
| | FAKD* | 34.63/0.9290 | 30.51/0.8453 | 29.21/0.8079 | 28.62/0.8612 |
| | CrossKD | 34.65/0.9290 | 30.50/0.8449 | 29.21/0.8079 | 28.60/0.8610 |
| | **AugKD** | **34.74/0.9296** | **30.54/0.8458** | **29.25/0.8088** | **28.79/0.8646** |
| ×4 | Scratch | 32.31/0.8966 | 28.69/0.7842 | 27.64/0.7384 | 26.37/0.7949 |
| | KD | 32.45/0.8980 | 28.76/0.7860 | 27.67/0.7400 | 26.49/0.7980 |
| | FitNet | 31.99/0.8899 | 28.50/0.7789 | 27.55/0.7353 | 25.90/0.7791 |
| | AT | 32.31/0.8967 | 28.69/0.7839 | 27.64/0.7385 | 26.29/0.7927 |
| | RKD | 32.39/0.8974 | 28.74/0.7856 | 27.67/0.7399 | 26.47/0.7981 |
| | FAKD* | 32.46/0.8980 | 28.77/0.7860 | 27.68/0.7400 | 26.50/0.7980 |
| | CrossKD | 32.46/0.8984 | 28.79/0.7863 | 27.69/0.7405 | 26.52/0.7992 |
| | **AugKD** | **32.56/0.8990** | **28.83/0.7870** | **27.72/0.7410** | **26.62/0.8020** |

between SR and other CV tasks. (**3**) The proposed AugKD method consistently outperforms the existing KD methods in all experimental settings by a large margin. For example, compared with the response-based KD method, the average PSNR improvements for the three types of networks on the Urban100 test set over three SR scales are 0.43 dB, 0.31 dB, 0.31 dB, respectively. Most existed KD methods are inapplicable to the transformer architecture network, but AugKD, as a response-based KD method, is able to compress the SwinIR model while exhibiting great performance.

**AugKD facilitates the student model to mimic the teacher model.** In Figure 2, we compare the effect of different KD methods by comparing the similarity of students' output and teacher's on the training and Urban100 testing sets, to evaluate how well the student learns to mimic the teacher model. It shows that AugKD makes the student not only effectively fit the teacher model on the training set but also imitate it on the test sets so that it can generalize better.

Table 4: The results of heterogeneous distillation using AugKD on the ×4 scale RCAN model.

| Teacher | BSD100 PSNR/SSIM | Urban100 PSNR/SSIM |
|---|---|---|
| (Scratch) | 27.64/0.7384 | 26.37/0.7949 |
| EDSR | 27.71/0.7406 | 26.59/0.8014 |
| SwinIR | 27.72/0.7408 | 26.59/0.8007 |

Table 5: NIQE scores on several real-world SR testing datasets. The lower, the better. Visual comparisons are provided in the appendix.

| Method | #Params | RealSR | DRealSR | OST300 |
|---|---|---|---|---|
| Scratch | 11.9M | 4.771 | 4.847 | 2.932 |
| Scratch | | 5.810 | 5.757 | 3.788 |
| KD | 1.24M | 5.425 | 5.408 | 3.652 |
| AugKD | | 5.398 | 5.378 | 3.494 |

**Experiment Results on Heterogeneous Settings.** We extend the experiments to heterogeneous settings where the teacher and student models have different network architectures, as presented in Table 4. Conventional feature-based KD or self-distillation methods are inapplicable to the cross-architecture setting, while AugKD can still effectively improve the student models. For instance,

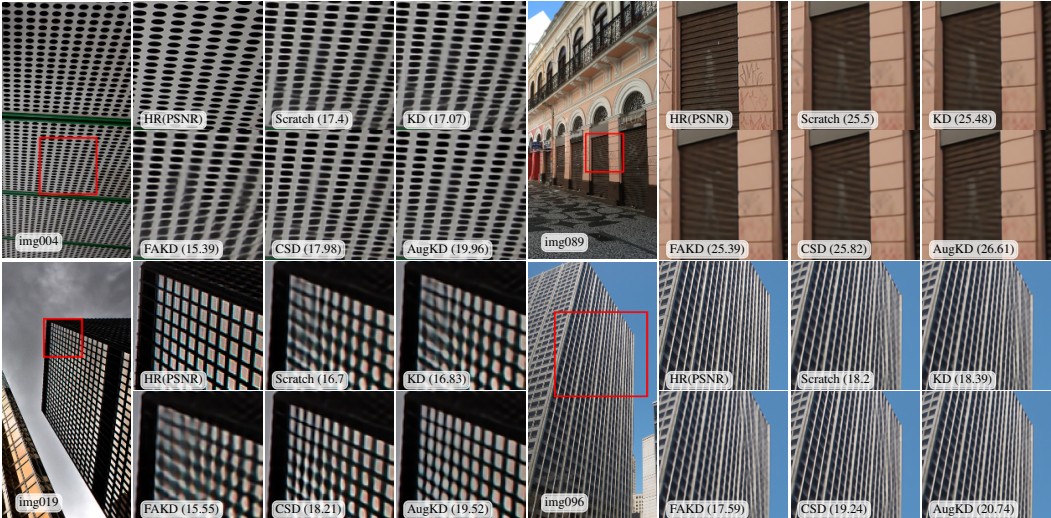

Figure 5: The ×4 SR examples of EDSR models on img004, img019, img089 and img096 from Urban100. PSNRs (dB) of the cropped regions are annotated below each image.

compared to the RCAN model trained from scratch, utilizing AugKD with an EDSR or SwinIR teacher model yields an increase in PSNR by 0.22dB at ×4 scale on Urban100 test set.

**Visual Comparison.** In Figure 5, we compare the visual quality of output images of the ×4 EDSR model trained by AugKD and other methods. To underscore the differences in detailed pattern and texture reconstruction, we took relatively small cropped portions and computed local PSNR metrics. Generally, a higher PSNR aligns with superior visual effect. For the reconstruction of textures (e.g. lines, edges, and complex patterns), the model trained with AugKD yields outputs that are both sharper and more similar to HR, indicating the superiority of AugKD.

**Experiment Results on Real-world SR task.**

To evaluate the performance of AugKD in real-world SR, we continue training the PSNR-oriented student SwinIR models at ×4 scale using the BSRGAN degradation model (Zhang et al., 2021a; Liang et al., 2021) on the DF2K dataset. The models are tested on three benchmark datasets: RealSR (Cai et al., 2019), DRealSR (Wei et al., 2020), and OST300 (Wang et al., 2018). The non-reference image quality assessment (NIQE) scores (Mittal et al., 2012) are presented in Table 5. The model trained with AugKD achieves lower NIQE scores and produces output images with more visually pleasing results, as shown in the supplementary material.

## 4.3 ABLATION ANALYSIS

**Impact of auxiliary distillation samples and label consistency regularization.** Table 6 shows the effect of the presented two modules, using EDSR baseline model (#Channel=64, #Block=16) distilled by our student model (#Channel=64, #Block=32). Further, Table 7 ablates the zoom-in ⊕ and zoom-out ⊖ operations. The result shows that adopting auxiliary distillation samples and label consistency regularization could lead to significant performance improvement upon logits-KD, whether used separately or together. For example, simply introduce the auxiliary images by zoom-in draws 0.31dB PSNR increment on Urban100 test set, and adding zoom-out and label consistency regularization yields an additional 0.16dB improvement.

**Integrate AugKD into other model compression methods.** We integrate AugKD with a SOTA SR network quantization method, Distribution-Aware Quantization (DAQ) (Hong et al., 2022), and use the full-precision model to supervise the quantized ones. Figure 6 shows the PSNR of quantized ×4 scale EDSR baseline models trained with and without KD, and the full results are provided in supplementary. It shows that vanilla Logits-KD has barely effects on the quantization, while AugKD could improve the quantized model by a large margin. We also integrate AugKD with the FAKD

Table 6: Ablation study of auxiliary distillation samples and label consistency regularization.

| Auxiliary samples | Label consistency | Urban100 |
|---|---|---|
| | | PSNR/SSIM |
| ✗ | ✗ | 24.87 / 0.7431 |
| ✓ | ✗ | 25.20 / 0.7558 |
| ✓ | ✓ | 25.34 / 0.7609 |

Table 7: Ablation study of the zoom in and zoom out operations.

| Zoom In ⊕ | Zoom Out ⊖ | Urban100 |
|---|---|---|
| | | PSNR / SSIM |
| ✗ | ✗ | 24.87 / 0.7431 |
| ✓ | ✗ | 25.18 / 0.7551 |
| ✗ | ✓ | 25.18 / 0.7552 |
| ✓ | ✓ | 25.20 / 0.7558 |

method in Table 8. The resulting models outperform the ones trained by FAKD greatly. The results indicate that AugKD can be effectively aggregated with other model compression techniques.

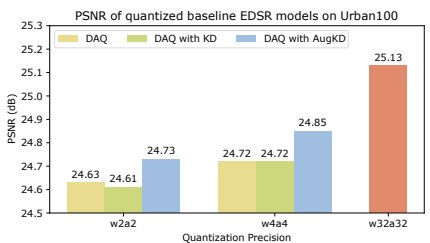

Figure 6: PSNR of quantized baseline EDSR model trained with and without KD.

Table 8: Experiment results of combining AugKD and FAKD.

| Model | Method | Urban100 |
|---|---|---|
| | | PSNR / SSIM |
| EDSR | Logits KD | 26.21 / 0.7897 |
| | FAKD | 26.18 / 0.7895 |
| | FAKD+AugKD | 26.30 / 0.7930 |
| | AugKD | 26.45 / 0.7966 |

Table 9: Comparison with data expansion. DF2K denotes DIV2K+Flickr2K.

| Training set | #Images | Training steps | Method | BSD100 | Urban100 |
|---|---|---|---|---|---|
| | | | | PSNR/SSIM | PSNR/SSIM |
| DIV2K | 800 | $2.5 \times 10^5$ | Scratch | 27.57/0.7356 | 25.94/0.7809 |
| | | | AugKD | **27.68/0.7390** | **26.32/0.7927** |
| DF2K | 3450 | $5 \times 10^5$ | Scratch | 27.62/0.7372 | 26.15/0.7872 |
| | | | KD | 27.67/0.7390 | 26.31/0.7925 |

**Comparison with data expansion** The proposed AugKD generates auxiliary distillation samples by simple data augmentations. Comparing with using a parametric generator or introducing additional training image data sources, it's more efficient and able to keep the training data have similar inputs. Table 9 compares AugKD with training or distilling with expanded data. We train the ×4 scale EDSR models on a much larger dataset (DF2K: DIV2K+Flickr2K (Timofte et al., 2017) with 3450 images). The number of iterations is doubled for the larger training set since the previous configuration ($2.5 \times 10^5$) is insufficient for the models to converge. Except that the ×4 SR networks are not initialized with the ×2 ones in this comparison, the other settings of the training recipe are the same. AugKD is superior to training with more input data in terms of both efficiency and performance.

## 5 CONCLUSION

In this work, we investigated the issues existing in KD for SR networks. Motivated by the findings, we present AugKD, a simple yet significant KD framework for SR, which outperforms existing methods and is applicable to a wide array of network architecture and SR tasks. Central to our approach is the auxiliary distillation samples generated by zooming augmentations, which facilitates the knowledge transfer from teacher to student model. Besides, we realize label consistency regularization in KD for SR, which further bolsters the student model's generalization capabilities. Extensive experiments are conducted across various SR tasks, benchmark datasets and diverse network backbones, consistently showing the out-performance of AugKD and endorsing its robust and effective.

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

## A    Supplymentary experiment results

### A.1    Training details

The SR models are trained using 800 images from the DIV2K dataset (Timofte et al., 2017) and evaluated on four benchmark datasets: Set5 (Bevilacqua et al., 2012), Set14 (Zeyde et al., 2012), BSD100 (Martin et al., 2001), and Urban100 (Huang et al., 2015). The low-resolution (LR) images used for training are generated by down-sampling the high-resolution (HR) images using bicubic degradation. The ×4 scale SR models are initialized from the corresponding ×2 scale models. During training, the input LR images are randomly cropped into $48 \times 48$ patches and augmented with random horizontal and vertical flips and rotations. For the FAKD and CSD methods, we follow the hyperparameter settings specified in the original papers and train the models ourselves if checkpoints are not provided, as noted in the results table. The zoom-in 🔍 operation for AugKD is performed by randomly cropping for efficiency. The zoom-out 🔍 operation is skipped during training for SwinIR, as the $I_{LR_{zo}}$ would be too small to serve as valid input to the model. The models are trained using the Adam optimizer (Kingma & Ba, 2014) with $\beta_1 = 0.9$, $\beta_2 = 0.99$, and $\epsilon = 10^{-8}$, with a batch size of 16 and a total of $2.5 \times 10^5$ updates. The initial learning rate is set to $10^{-4}$ and decays by a factor of 10 every $10^5$ iterations. The proposed KD method is implemented using the BasicSR framework (Wang et al., 2022a) and PyTorch 1.10, with training performed on 4 NVIDIA V100 GPUs.

### A.2    Experiment results of SwinIR network

We compare AugKD with other applicable KD methods on distilling transformer-based SR model, SwinIR. The result shows the superiority and universality of AugKD.

Table 10: Quantitative comparison (average PSNR/SSIM) between AugKD and other applicable distillation methods for **SwinIR** of three SR scales. Best performance is highlighted in bold.

| Scale | Method | Set5 PSNR/SSIM | Set14 PSNR/SSIM | BSD100 PSNR/SSIM | Urban100 PSNR/SSIM |
|---|---|---|---|---|---|
| ×2 | Scratch | 38.01/0.9607 | 33.57/0.9178 | 32.19/0.9000 | 32.05/0.9279 |
| | KD | 38.04/0.9608 | 33.61/0.9184 | 32.22/0.9003 | 32.09/0.9282 |
| | **AugKD** | **38.13/0.9610** | **33.78/0.9194** | **32.26/0.9007** | **32.63/0.9327** |
| ×3 | Scratch | 34.41/0.9273 | 30.43/0.8437 | 29.12/0.8062 | 28.20/0.8537 |
| | KD | 34.44/0.9275 | 30.45/0.8443 | 29.14/0.8066 | 28.23/0.8545 |
| | **AugKD** | **34.55/0.9285** | **30.53/0.8456** | **29.20/0.8080** | **28.53/0.8604** |
| ×4 | Scratch | 32.31/0.8955 | 28.67/0.7833 | 27.61/0.7379 | 26.15/0.7884 |
| | KD | 32.27/0.8954 | 28.67/0.7833 | 27.62/0.7380 | 26.15/0.7887 |
| | **AugKD** | **32.41/0.8973** | **28.79/0.7860** | **27.69/0.7405** | **26.43/0.7972** |

### A.3    Comparison of training costs

As shown in Table 11, AugKD outperforms Logits-KD by **0.55dB** PSNR with an increase of only 0.21s training time per step. Considering the significant performance gains from AugKD, the extra cost on training time is mild and acceptable.

Table 11: Training expenses of KD methods for ×2 SR on EDSR.

| KD methods | KD | FitNet | FAKD | CSD | AugKD |
|---|---|---|---|---|---|
| Time (s/step) | 0.49 | 0.56 | 0.56 | 1.18 | 0.70 |
| Urban100 PSNR | 31.98 | 30.46 | 32.04 | 32.26 | 32.53 |

### A.4    Teacher models' results

In this work, we use EDSR, RCAN, and SwinIR as backbone models for the experiments. Their specifications and statistics are provided in Table 1. We use the public checkpoints of these teacher models for distilling the student models, with quantitative results summarized in Table 12.

Table 12: Quantitative results of teacher models

| Scale | Model | Set5 PSNR/SSIM | Set14 PSNR/SSIM | BSD100 PSNR/SSIM | Urban100 PSNR/SSIM |
|---|---|---|---|---|---|
| ×2 | EDSR | 38.20/0.9610 | 34.02/0.9204 | 32.37/0.9018 | 33.10/0.9363 |
| | RCAN | 38.27/0.9614 | 34.13/0.9216 | 32.39/0.9024 | 33.18/0.9371 |
| | SwinIR | 38.36/0.9620 | 34.14/0.9227 | 32.45/0.9030 | 33.40/0.9394 |
| ×3 | EDSR | 34.76/0.9290 | 30.66/0.8481 | 29.32/0.8104 | 29.02/0.8685 |
| | RCAN | 34.74/0.9299 | 30.65/0.8482 | 29.32/0.8111 | 29.09/0.8702 |
| | SwinIR | 34.89/0.9312 | 30.77/0.8503 | 29.37/0.8124 | 29.29/0.8744 |
| ×4 | EDSR | 32.65/0.9005 | 28.95/0.7903 | 27.81/0.7440 | 26.87/0.8086 |
| | RCAN | 32.64/0.9000 | 28.85/0.7890 | 27.74/0.7430 | 26.75/0.8070 |
| | SwinIR | 32.72/0.9021 | 28.94/0.7914 | 27.83/0.7459 | 27.07/0.8164 |

## B   MORE VISUAL RESULTS

In Figure 7, we present additional visual comparisons of AugKD with other KD methods on Urban100. AugKD restores more structural details and reduces blurring artifacts. The AugKD was evaluated on the real-world SR task in Table 5, where it outperformed the baselines across several testing datasets. In Figure 8, we show visual comparisons for real-world SR, further highlighting its superior performance.

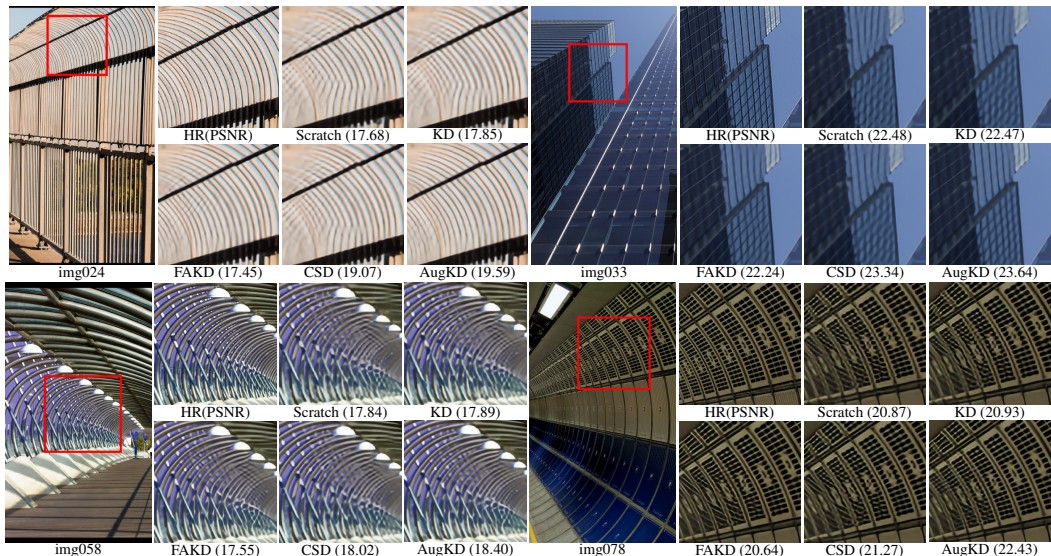

Figure 7: The ×4 super resolution results of EDSR models on image 033, 078, 058 and 024 from Urban100. PSNRs (dB) of the cropped regions are annotated below each image. Zoom in 🔍 for the best view.

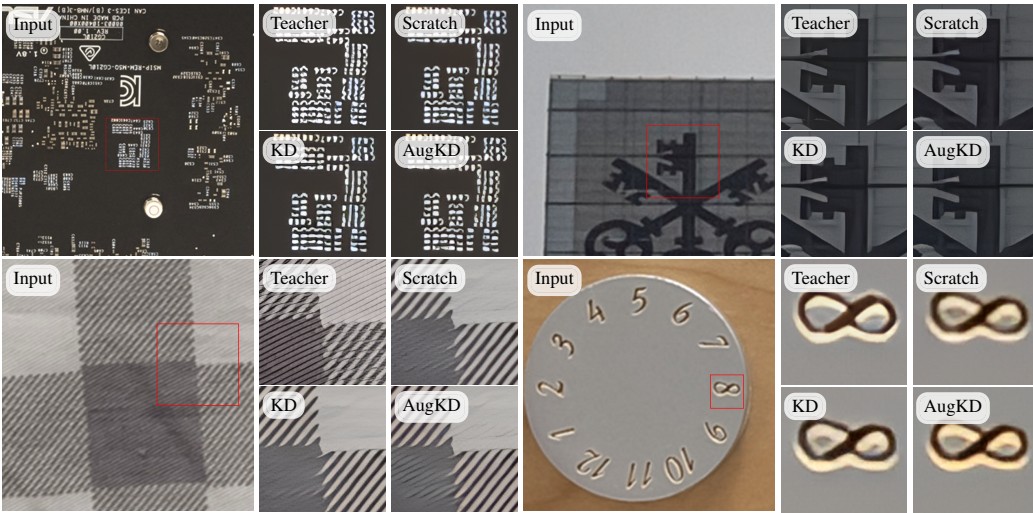

Figure 8: Visual comparisons of representative real-world SR images at ×4 SR scale. AugKD outperforms other methods in artifact removal and detail restoration, producing outputs more similar to the teacher model. Zoom in 🔍 for optimal viewing.

