# OpenReview forum: "AugKD: Ingenious Augmentations Empower Knowledge Distillation for Image Super-Resolution"
_ICLR.cc/2025/Conference — ICLR 2025 Poster_

### Official Review · Reviewer_3UKt · 2024-10-29

**Soundness:** 2
**Presentation:** 4
**Contribution:** 3
**Rating:** 6
**Confidence:** 4

**Summary:**

This paper proposes AugKD, an innovative knowledge distillation (KD) technique specifically designed for image super-resolution (SR). AugKD incorporates zooming augmentations and label consistency regularization to enhance the training process. In this approach, both randomly cropped high-resolution (HR) patches and their down-sampled low-resolution (LR) counterparts are fed into a pre-trainedd teacher model to generate target labels. These labels are then used to guide the training of the student model. To further improve the robustness and generalization of the student model, consistency regularization is applied through a series of invertible data augmentations. Extensive experiments have been conducted across multiple public image super-resolution datasets, demonstrating the effectiveness and versatility of the proposed method.

**Strengths:**

1. A novel KD method is proposed in this paper. AugKD improves the training process by using zooming augmentations and label consistency regularization. To make the student model more robust and versatile, consistency regularization is applied using a series of invertible data augmentations. Extensive quantitative experiments and qualitative analysis are provided to demonstrate the validity of the methodology
2. Compared with previous methods, the performance is improved on models with multiple scales. For instance, compared with training from scratch, the performance of RCAN on x4 scale is improved by 0.25dB on Urban dataset.
3. AugKD is general and effective, easy to follow, and convenient for reproducing the method.
4. Paper is well written and organized.

**Weaknesses:**

1. This paper proposed multiple effective improvements, while I'm curious that, besides zooming augmentations, could other data augmentation methods improve performance?
2. The ablation of the label consistency is not sufficient. Have the authors tried other non-invertible ways of regularization?

**Questions:**

1. Referred to Tab.9 in the paper, could you explain the reason why the performance of the combination of FAKD and AugKD is lower than AugKD?
2. The performance of AugKD on the X4 scale RCAN model is presented in Tab.3, why is it better than the result of heterogeneous distillation in Tab.5?
3. Does $L_{dukd}$ in Fig. 1 indicate the same with the $L_{augkd}$ computed in Fig.4?

---

> ### Author Response · Authors · 2024-11-23
>
> We greatly appreciate your valuable review comments . Below is our point-by-point response to your concerns and questions. Please let us know if there's anything we can clarify further.
>
> # Weakness 1: Potential of other data augmentations in AugKD.
>
> Reason for adopting zooming augmentations: AugKD identifies and addresses a key challenge in KD for SR that the teacher model’s function of transferring knowledge to the student is overshadowed by the ground truth labels, as the teacher’s outputs are noisy approximations of high-resolution images. By introducing auxiliary distillation samples, AugKD allows the student model to extract meaningful prior knowledge from the teacher model’s outputs.  The zoom-in and zoom-out augmentations were specifically chosen because they are well-aligned with SR tasks, generating images closely related to the training data distribution and avoiding distributional shifts. They also allow the student to learn not only from direct reconstructions but also from implicit task-relevant patterns, such as fine-grained textures (zoom-in) and broader spatial consistency (zoom-out).
>
> Other data augmentations cannot effectively address the challenges in KD for SR, and might introduce distribution shifts to the training data. Hence, their effects on improving the student SR models are limited.
>
> # Weakness 2: Regularization with non-invertible augmentations
>
> The non-invertible augmentations, such as cropping or translation, would fundamentally alter the spatial correspondences between teacher and student models' output and make them not comparable. And since the LR images are bicubic degradation of corresponding HR labels in training of SR,  the augmentations like noising and blurring are not strongly related with the task and would making images distributed different from the training data. Thus, we prioritize invertible augmentations like flipping and rotation, which can preserve the spatial integrity of input images.
>
> Nevertheless, we recognize the importance of exploring alternative augmentation and regularization strategies. While non-invertible augmentations may not be directly compatible with SR task's requirements, for other low-level CV tasks like image denosing and debluring, the label consistency regularization can be realized by passing differently noised or blurred images to the student and teacher model.
>
> # Question 1: Combination of AugKD and FAKD
>
> The result in Tab.9 indicates that the introduction the auxiliary distillation samples and label consistency regularization into FAKD can significantly improve the student model's performance. The performance of the combined method (FAKD + AugKD) being slightly worse than AugKD alone can be attributed to two factors. First, the FAKD is detrimental for the student model, as shown in the main experiment results. And the hyper-parameters for the combined method were not specifically tuned to optimize the balance between FAKD's feature matching modules and AugKD, potentially leading to suboptimal results.
>
> # Question 2: Heterogeneous distillation experiments
>
> The superior performance of AugKD on the X4 scale RCAN model in Tab. 3 compared to the heterogeneous distillation results in Tab. 5 is primarily due to the architectural alignment in homogeneous distillation. In Tab. 3, both the teacher and student models share the RCAN architecture, which facilitates effective transfer of task and network specific priors. In contrast, heterogeneous distillation (Tab. 5) involves different architectures for the teacher (e.g., EDSR or SwinIR), which can introduce challenges in knowledge transfer due to mismatched architectural representations. Furthermore, the large capacity gap between models may lead to suboptimal student performance, as shown in Table 11 of the supplementary material, where the EDSR and SwinIR teachers significantly outperform the RCAN teacher in isolation. These factors collectively explain the observed performance difference.
>
> # Question 3: Issues on the typos
>
> Thank you for pointing out the misleading typo in the manuscript. The two notations denotes the same loss term. We will carefully revise the manuscript to reduce the reader's confusion.

---

### Official Review · Reviewer_HXWn · 2024-11-03

**Soundness:** 2
**Presentation:** 3
**Contribution:** 3
**Rating:** 6
**Confidence:** 4

**Summary:**

This paper introduces AugKD, a novel method aimed at improving image super-resolution (SR) by leveraging data augmentations to generate auxiliary distillation samples and enforce consistency regularization. This work analyzes the mechanisms of KD for SR and propose AugKD adapted to the unique task with label consistency regularization. Extensive experiments on various SR tasks are presented across multiple datasets to validate the proposed approach.

**Strengths:**

1. The paper thoroughly analyzes the mechanics and distinct challenges of knowledge distillation (KD) in the context of SR, proposing the use of data augmentations to enhance distillation.
2. The AugKD strategy is adaptable to different SR models and tasks, yielding substantial performance improvements across several networks and settings.
3. The well-organized structure and clearly described method facilitates reproducibility.

**Weaknesses:**

1. The visualization in Fig. 2 is unclear. Replacing it with a bar plot may improve readability and convey the idea more effectively.
2. Although lines 241-244 highlight that the adaptive selection of zoom-in samples is ineffective, it lacks sufficient experiments to support this claim.
3. The motivation by using label consistency regularization is unclear.

**Questions:**

1. How does AugKD affect training efficiency in comparison to other KD techniques?
2. What's the rationale behind the specific choice of zoom-in and zoom-out augmentations for AugKD?
3. Given that some other image augmentations, such as translation, are also invertible, would it be beneficial to include them in the consistency regularization module?
4. What is the motivation of using label consistency regularization for SR? Is it also also suitable for other low-level tasks?
5. Does the proposed method also adapt to other backbones, like Mamba?

---

> ### Author Response · Authors · 2024-11-23
>
> We greatly appreciate your valuable review comments . Below is our point-by-point response to your concerns and questions. Please let us know if there's anything we can clarify further.
>
> # Weakness 1: Unclear visualization in Figure 2.
>
> Thank you for your suggestion regarding Figure 2. To improve the visualization for better readability, we will replace the figure with a bar chart to clearly present the differences in PSNR values between training methods.
>
> # Weakness 2: Issues on adaptive zoom-in sample selection
>
> In our exploratory experiments, we explored and compared several auxiliary distillation sample generation strategies: 1) adaptively select the zoom-in samples based on reconstruction difficulty, indicated by the PSNR between the teacher model’s output and the ground-truth HR labels, 2) equally split the HR into several grids and use them all as auxiliary distillation samples and 3) randomly pick an auxiliary distillation sample. The result below shows that the random selection of zoom-in sample slightly outperforms the alternative approaches. And given the increased computational overhead required for adaptive selection and training with more auxiliary distillation samples, random selection remains a more efficient and practical option.
>
> | Method               | Set5             | Set14            | BSD100           | Urban100         |
> | --------------------------------------------------- | ---------------- | ---------------- | ---------------- | ---------------- |
> | Student (Scratch)                                   | 31.13/0.8783     | 27.94/0.7664     | 27.12/0.7216     | 28.87/0.7432     |
> | KD                                                  | 31.16/0.8791     | 27.95/0.7670     | 27.13/0.7223     | 24.87/0.7431     |
> | AugKD-v1: (Zoom-in at the most difficult patch)    | 31.47/0.8843     | 28.18/0.7719     | 27.27/0.7263     | 25.13/0.7553     |
> | AugKD-v2: (Zoom-in at all equally grids)           | 31.51/0.8849     | 28.18/0.7722     | 27.28/0.7266     | 25.15/0.7541     |
> | **AugKD (randomly select $I\_{\text{HR}\_{zi}}$)** | **31.51/0.8849** | **28.19/0.7726** | **27.29/0.7268** | **25.18/0.7551** |
>
> We will include these experimental results in the supplementary material to provide evidence supporting our claim and to clarify the rationale behind randomly zoom-in sample selection.
>
> # Weakness 3 and Question 4: Motivation and applicability for label consistency regularization
>
> Label consistency regularization is motivated by the need to improve the robustness and generalization capability of the student model in KD. Normal KD matches the student and teacher models on the same input images, while a student model with strong robustness and generalizability should be able to maintain such match on augmented inputs. By enforcing consistency between the outputs of the student model for augmented input and the unaltered outputs of the teacher model, the student is encouraged to learn invariant representations that align with the teacher’s knowledge, irrespective of input perturbations. This mechanism makes the student expose to diverse data distributions, and guides it to focus on and learn SR task-related information that remains unaffected by input perturbations.
>
> The label consistency regularization is not specific to SR but is applicable to other low-level tasks such as denoising, deblurring, and image enhancement. These tasks share the characteristic of pixel-level input-output dependencies, and maintaining teacher-student consistency under transformations is crucial. By enforcing invariance to input variations, this regularization technique can improve the robustness and effectiveness of models for various low-level tasks.

---

> ### Author Response · Authors · 2024-11-23
>
> # Question 1: Training efficiency of AugKD
>
> As shown in the comparison on X2 EDSR, the AugKD outperforms Logits-KD by 0.55dB PSNR with an increase of only 0.21s training time per step. Considering the significant performance gains in test time, the mild extra cost in training is acceptable.
>
> | KD methods    | KD    | FitNet | FAKD  | CSD   | AugKD |
> | ------------- | ----- | ------ | ----- | ----- | ----- |
> | Time (s/step) | 0.49  | 0.56   | 0.56  | 1.18  | 0.70  |
> | Urban100 PSNR | 31.98 | 30.46  | 32.04 | 32.26 | 32.53 |
>
> # Question 2: Rationale behind the choice zoom-in and zoom-out augmentations
>
> In knowledge distillation for super-resolution models, the teacher model's intended function, transferring knowledge to student, is shaded by the ground truth HR labels, due to the nature that the teacher model's outputs are noisy approximation to the high-resolution images. So there is barely "dark knowledge" such as the inter-class relationship or label smoothing in the classification task. The AugKD addresses the issue and re-functions the teacher model by matching student and teacher models on the auxiliary samples that are generated efficiently and distribute closely with original training data.
>
> The zoom-in and zoom-out augmentations were specifically chosen because they are well-aligned with the goals of auxiliary distillation sample generation in SR. Besides, they also allow the student to learn not only from direct reconstructions but also from implicit task-relevant patterns, such as fine-grained textures (zoom-in) and broader spatial consistency (zoom-out).
>
> # Question 3: Alternative inverse data augmentations for label consistency regularization
>
> While translation is indeed an invertible augmentation, it is not included in the consistency regularization module because it does not align well with the pixel-level nature of SR. Translation introduces disjointed artifacts at image boundaries, which are uncommon during testing and might mislead the student model during training.
>
> In the training of SR model, maintaining precise spatial correspondences in input and output is critical for accurately reconstructing high-resolution images. Mild translation can disrupt this correspondence, introducing inconsistencies that would negatively impact the learning process. Thus, we prioritize augmentations like flipping and rotation, which preserve the spatial integrity required for SR tasks.
>
> # Question 5: Applicability to other SR network backbones
>
> The proposed AugKD method is logits-based and is not constrained by specific network architectures. This flexibility allows it to be applied to various SR backbones, including Mamba and other architectures.
>
> To further demonstrate its applicability to various backbones, we provide the experiment results on a latest SR network, DRCT[1] below. We use the large version of DRCT as the teacher model and distill the DRCT SR network by different KD methods.
>
> | Method    | X4 DRCT Urban100 PSNR/SSIM |
> | --------- | ----------------------------- |
> | Teacher   | 28.70/0.8508                  |
> | Student   | 27.23/0.8188                  |
> | Logits-KD | 27.22/0.8181                  |
> | FitNet    | 27.21/0.8177                  |
> | AugKD     | 27.43/0.8226                  |
>
> AugKD consistently outperforms previous SR KD methods on different backbones.
>
> [1] Hsu, C. C., Lee, C. M., & Chou, Y. S. (2024). DRCT: Saving Image Super-resolution away from Information Bottleneck. arXiv:2404.00722.

---

> > ### Comment · Reviewer_HXWn · 2024-11-26
> > **rebuttal**
> >
> > Thank you for providing the detailed rebuttals. All my comments have been properly addressed. The contributions on the data augmentations and label consistency regularization are solid, which have been effectively evaluated through comprehensive experiments. Thus, I tend to accept it.

---

> > > ### Author Response · Authors · 2024-11-26
> > >
> > > Thank you for your valuable feedback and constructive comments. We greatly appreciate the time and attention you have dedicated to this process.

---

### Official Review · Reviewer_oSVc · 2024-11-07

**Soundness:** 2
**Presentation:** 2
**Contribution:** 2
**Rating:** 6
**Confidence:** 3

**Summary:**

The paper explores an improved augmentation strategy for knowledge distillation (KD) specifically in image super-resolution (SR). It introduces AugKD, which uses unpaired data augmentations to create auxiliary distillation samples and enforce label consistency. This approach addresses limitations in traditional KD methods by enhancing the student model’s learning process through diverse training samples, aiming to improve efficiency and effectiveness in SR tasks.

**Strengths:**

- **Motivation**: The paper provides a clear motivation for improving data augmentation strategies in knowledge distillation (KD) for super-resolution (SR), highlighting the unique challenges in SR tasks.
- **Comprehensive Ablations**: Extensive ablation studies test various experimental setups for KD in SR, demonstrating a thorough examination of the method's effectiveness under different conditions.

**Weaknesses:**

- **Modest Improvements**: Results in Figure 2 and Tables 2–4 show only slight gains, questioning the practical value of AugKD over existing methods.
- **Limited Insight on Augmentation Impact**: Ablation studies explain augmentation effects but don’t clarify *why* this strategy improves KD. Further detail on what specific features AugKD captures would be helpful.
- **Augmentation Benefits Unclear**: It’s unclear how augmentations help representations learned through KD or why prior methods failed to capture these.
- **Potential Architecture Constraints**: While AugKD is claimed to be generalizable, performance with some architectures (like SwinIR) suggests possible limitations.

**Questions:**

- Please see weakness.

---

> ### Author Response · Authors · 2024-11-23
>
> We greatly appreciate your valuable review comments. Below is our point-by-point response to your concerns and questions. Please let us know if there's anything we can clarify further.
>
> # Weakness 1: Concerns about model performance
>
> Thank you for your feedback regarding the reported improvements. The improvements achieved by AugKD are consistent across all networks, datasets, and SR scales, demonstrating its robustness and practical value. These performance gains are not merely marginal but stem from the introduction of auxiliary distillation samples and label consistency regularization.  As shown in Table 3, on the Urban100 test set, AugKD improves the PSNR of RCAN networks over the strongest baseline KD method (the underlined results), by 0.17dB, 0.17dB, and 0.10dB on three SR scales. The PSNR improvements over training from scratch are 2 to 7 times larger than those achieved by the strongest baseline methods.
>
> In addition to performance improvements, AugKD offers distinct advantages in applicability and flexibility. It works effectively with both CNN-based and Transformer-based architectures, addressing compatibility limitations of many existing methods. Its versatility is further proved by its integration into diverse tasks, such as SR network quantization and real-world SR scenarios. These factors collectively demonstrate the practical value and broad potential of AugKD.

---

> ### Author Response · Authors · 2024-11-23
>
> # Weakness 2, 3: Motivations and insights of augmentations
>
> **Augmentation impacts of auxiliary distillation samples**:  AugKD identifies and addresses a key challenge in KD for SR: the teacher model’s function of transferring knowledge to the student is overshadowed by the ground truth labels, as the teacher’s outputs are noisy approximations of high-resolution images. So there is barely "dark knowledge" like the inter-class relationship or label smoothing in the classification task. By introducing auxiliary distillation samples generated by zoom-in and zoom-out augmentations, AugKD allows the student model to extract meaningful prior knowledge from the teacher model’s outputs. They also allow the student to learn not only from direct reconstructions but also from implicit task-relevant patterns, such as fine-grained textures (zoom-in) and broader spatial consistency (zoom-out), that are challenging to capture with standard KD methods.
>
> **Augmentation impacts of label consistency regularization**: the label consistency regularization further strengthens student model's generalization by enforcing consistent matches with teacher model across augmented views of the same input. It's motivated by the need to improve the robustness and generalization capability of the student model in KD. Normal KD matches the student and teacher models on the same input images, while a student model with strong robustness and generalizability should be able to maintain such alignment on augmented inputs. By enforcing consistency between the outputs of the student model for augmented input and the unaltered outputs of the teacher model, the student is encouraged to learn invariant representations that align with the teacher’s knowledge, irrespective of input perturbations. This mechanism makes the student expose to diverse data distributions, and guides it to focus on and learn SR task-related information that remains unaffected by input perturbations.
>
> Prior KD methods failed to fully utilize such task-related augmentations because most of them relied on static feature-based approaches, leading to the sensitivity to architectural mismatches or limitation to high-level vision tasks. While the AugKD identifies the issues in KD for SR and proposed augmentation-based solutions accordingly.

---

> ### Author Response · Authors · 2024-11-23
>
> # Weakness 4: Concern on architecture constraints
>
> In the experiment section, EDSR and RCAN were selected as benchmarks because they are widely used in existing KD studies for SR tasks. While many feature-based KD methods are specifically designed for CNN-based models, AugKD demonstrates its broad compatibility with both CNNs and Transformers, such as SwinIR, as it is logits-based and independent from specific network architectures.
>
> To provide a more comprehensive evaluation of SwinIR, we conducted additional experiments comparing AugKD with FitNet and FAKD on the X4 scale SwinIR model. The results below show that prior feature-based KD methods, such as FitNet and FAKD, fail to improve the SwinIR model effectively, whereas AugKD achieves superior performance:
>
> | Method | Set5         | Set14        | BSD100       | Urban100     |
> | ------ | ------------ | ------------ | ------------ | ------------ |
> | Logits-KD     | 32.27/0.8954 | 28.67/0.7833 | 27.62/0.7380 | 26.15/0.7887 |
> | FitNet | 32.08/0.8925 | 28.51/0.7800 | 27.53/0.7354 | 25.80/0.7779 |
> | FAKD   | 32.06/0.8926 | 28.52/0.7800 | 27.53/0.7354 | 25.81/0.7780 |
> | AugKD  | 32.41/0.8973 | 28.79/0.7860 | 27.69/0.7405 | 26.43/0.7972 |
>
> To further demonstrate its applicability and effectiveness for various architectures, we evaluated AugKD on a recent state-of-the-art SR network, DRCT [1]. Using the large version of DRCT as the teacher model, we distilled the X4 scale DRCT student network using different KD methods. The results confirm that AugKD provides notable improvements compared to other methods:
>
> | Method    |  Urban100 PSNR/SSIM |
> | --------- | ----------------------------- |
> | Teacher   | 28.70/0.8508                  |
> | Student   | 27.23/0.8188                  |
> | Logits-KD | 27.22/0.8181                  |
> | FitNet    | 27.21/0.8177                  |
> | AugKD     | 27.43/0.8226                  |
>
> Hope these results highlighting AugKD’s flexibility and effectiveness across diverse SR architectures, can address your concerns about potential architecture constraints.
>
> [1] Hsu, C. C., Lee, C. M., & Chou, Y. S. (2024). DRCT: Saving Image Super-resolution away from Information Bottleneck. arXiv:2404.00722.

---

> > ### Comment · Reviewer_oSVc · 2024-11-26
> > **Response to Reviewer Comments**
> >
> > Thanks for the detailed response. I think they mostly address my concerns on the performance gain and motivation for the designs.  I would like to raise my score to 6.

---

> > > ### Author Response · Authors · 2024-11-26
> > >
> > > Thank you for your valuable feedback and constructive comments. We greatly appreciate the time and attention you have dedicated to this process.

---

### Official Review · Reviewer_PKbV · 2024-11-07

**Soundness:** 3
**Presentation:** 2
**Contribution:** 2
**Rating:** 6
**Confidence:** 4

**Summary:**

The paper proposes AugKD, a new knowledge distillation method for image super-resolution. AugKD contains two special designs, auxiliary
distillation sample generation, and label consistency regularization. By comparing with other distillation counterparts on four benchmark datasets, AugKD shows its superiority.

**Strengths:**

1. The paper has a deep insight into super-resolution tasks and its method is simple and effective.
2. The code is also provided for reproduction.

**Weaknesses:**

1. **The paper writing could be further improved.** For instance, the authors claim the motivation - *'the teacher's output contains barely extra information exceeding GT, thus the “dark knowledge” of the teacher being hardly transferred to the student model through KD'* in Line #76-78 in the Introduction part. However, in Section 3.2 (Motivation) and Figure 2, the authors show the motivation by measuring the PSNR of outputs between the teacher and the student. It seems that the two statements are a little bit contradictory, as the former one indicates that the teacher's outputs can not be good learning materials but the second one leverages the the teacher's outputs as the reference for evaluating whether KD method is good or not. Such circumstances make it hard to understand the central idea of the paper.

2. **The paper lacks further deep analysis of where the performance gains are from.** From the results in Figure 2, it seems that the gains are from improving the fidelity between the teacher and the student. A further question is *Why AugKD can improve the fidelity?* And in Lines #521-529, the authors compare AugKD with data expansion. Thus, a question arises *Does the improvement of fidelity from the expansion of the training set by augmentation?* From another perspective, the question is *how does the augmentation strength affect the fidelity and the final distillation results?* By answering such a series of questions, the paper can help the readers understand the intrinsic mechanism of AugKD.

3. **A minor question about the design of the method.** Although I think the design of the inverse augmentation is clear and plausible, I'm still curious about what would happen if we dropped the inverse augmentation and added the augmentation at the end of the teacher's model in the training stage and still utilized the same architecture as the method in the inference stage.

**Questions:**

See Weakness.

---

> ### Author Response · Authors · 2024-11-23
>
> We greatly appreciate your valuable review comments. Below is our point-by-point response to your concerns and questions. Please let us know if there's anything we can clarify further.
>
> # Weakness 1: Motivation and the role of the teacher’s outputs.
>
> Thank you for pointing out the potential inconsistency between our statements in the introduction and the motivation section.
>
> The statement in the introduction highlights the limitations of directly aligning the student with the teacher’s output caused by the inherent noise in the teacher’s predictions in super-resolution tasks. This does not intend to suggest that the teacher’s outputs are entirely unsuitable as learning materials, but rather that they are not effectively used. When there is the GT label ($\mathcal{L}\_{rec}$ is available), matching student and teacher models' outputs ($\mathcal{L}\_{kd}$) cannot effectively transfer the teacher model's knowledge.
>
> Section 3.2 and Figure 2 illustrate that while existing KD methods fail to fully utilize the teacher model to guide the student, AugKD makes the teacher’s outputs to serve as valuable learning references and transfers teacher's knowledge to student model through auxiliary distillation samples ($\mathcal{L}\_{augkd}$). Therefore, the student model perform more similarly with the teacher model as indicated by the higher PSNR(S,T).
>
> We will revise the manuscript to explicitly connect these points, ensuring that our central motivation and methodology are more clearly articulated.
>
> # Weakness 2: Explanation for the performance gain of AugKD
>
> Thank you for the series of questions that lead us to reflect more deeply on the mechanics of the AugKD.
>
> The performance improvements of AugKD stem primarily from the combination of auxiliary distillation samples and label consistency regularization. AugKD identifies and addresses a key challenge in KD for SR: the teacher model’s function of transferring knowledge to the student is overshadowed by the ground truth labels, as the teacher’s outputs are noisy approximations of high-resolution images.
>
> By introducing auxiliary distillation samples, AugKD allows the student model to extract meaningful prior knowledge from the teacher model’s outputs, enhancing the fidelity between the teacher and student models as a by-product. Label consistency regularization further strengthens generalization by enforcing consistent outputs across augmented views of the same input, which helps the student model adapt to diverse inputs.
>
> While data expansion increases the training set size, it does not address the issue of knowledge transfer bottleneck. Simply expanding the data cannot mitigate the shading effect of ground truth labels on the teacher model’s outputs. AugKD, by contrast, is explicitly designed to extract and utilize the teacher’s knowledge effectively, achieving performance gains beyond what data expansion alone could provide.
>
> Regarding augmentation strength in the label consistency regularization module, we have conducted exploratory experiments on it and the results are supplemented below.  In the implementation of label consistency regularization, we independently applied four invertible augmentations (invert color, horizontal flip, vertical flip, and transpose) to $I\_{{LR}\_{zo}}$ and $I\_{{HR}\_{zi}}$  with a probability parameter  $p\_{\text{aug}}$ . The results below illustrate the impact of  $p\_{\text{aug}}$  on performance:
>
> | Aug strength hyper-parameter $p\_{\text{aug}}$ | Set5          | Set14         | BSD100        | Urban100      |
> | ------------------------------------------------------ | ------------- | ------------- | ------------- | ------------- |
> | 0.3                                                    | 31.536/0.8854 | 28.207/0.7727 | 27.291/0.7269 | 25.185/0.7553 |
> | 0.5                                                    | 31.513/0.8850 | 28.198/0.7725 | 27.289/0.7267 | 25.181/0.7553 |
> | 0.7                                                    | 31.523/0.8853 | 28.203/0.7725 | 27.291/0.7267 | 25.185/0.7553 |
>
> The results indicate that an optimal augmentation strength ( $p\_{\text{aug}} = 0.3$ ) gives the best performance of student model, and this strength parameter was used in our experiments. We will include these results and findings in the revised manuscript to clarify the mechanism and impact of AugKD.
>
> # Weakness 3: Design of label consistency regularization.
>
> In the current implementation, invertible data augmentations are applied to the input and output of the student model while leaving the teacher model unchanged. Dropping the inverse augmentation and instead applying the same augmentation to the teacher model’s output would indeed be mathematically equivalent.

---

> > ### Comment · Reviewer_PKbV · 2024-11-24
> > **Reply to the Authors**
> >
> > Thank you for your response, which addresses most of my concerns. I have updated my score from 5 to 6.

---

> > > ### Author Response · Authors · 2024-11-24
> > >
> > > We sincerely thank you for your support and positive evaluation!

---

### Official Review · Reviewer_bsJu · 2024-11-08

**Soundness:** 3
**Presentation:** 3
**Contribution:** 2
**Rating:** 6
**Confidence:** 3

**Summary:**

This paper proposes a new method for knowledge distillation for image super-resolution. The authors propose using auxiliary training samples by zoom in and zoom out operations on the training images, and apply label consistency regularization by data augmentation and inverse augmentation.

**Strengths:**

The motivation makes sense that for image super-resolution knowledge distillation, the guidance of the teacher model is shaded by the ground truth. So the authors propose a specific knowledge distillation training paradigm for image super-resolution.
Experiments show that the proposed method surpasses scratch training and 7 baseline knowledge distillation methods.
The ablation studies verifies that the proposed auxiliary distillation samples and label consistency regularization improve student model performance.

**Weaknesses:**

The question answered by the paper is not a major one, as it is a knowledge distillation method specifically for the image super-resolution task. Does it also apply to other low-level tasks?
The image super-resolution models used for experiments are not state-of-the-art. EDSR is from 2017 and RCAN is from 2018. SwinIR is newer from 2021 but only "Scratch" and "KD" is compared with the proposed method for SwinIR. As the proposed method is claimed to be model-agnostic, it is supposed to be applied to more advanced models to demonstrate the effectiveness.

**Questions:**

For the zoom out operation, we have the ground truth for I_LR_zo, and given the analysis in Section 3.2, the teacher model output for I_LR_zo would be shaded by the ground truth. So there seems to be additional complexity for this path to go through the teacher model.

---

> ### Author Response · Authors · 2024-11-23
>
> We greatly appreciate your valuable review comments. Below is our point-by-point response to your concerns and questions. Please let us know if there's anything we can clarify further.
>
> # Weakness 1: Applicability of AugKD
>
> The idea of AugKD can indeed be extended to other low-level vision tasks by adapting the auxiliary distillation sample generation to fit task-specific data requirements. Since AugKD addresses a common issue in the KD for these pixel level CV tasks. For instance, when distill denoising or de-blurring models, task-related augmenting operations can be employed to generate auxiliary samples tailored for effective knowledge transfer. Besides, the invertible data augmentations can be directly plugged in these tasks to realize label consistency regularization. We will incorporate more examples and discussions in the manuscript and clarify this aspect further.
>
> # Weakness 2: Experiments on more advanced SR models
>
> The EDSR and RCAN models were selected because they are widely adopted benchmarks in existing super-resolution knowledge distillation studies. While many feature-based distillation methods are specifically tailored to CNN-based models, AugKD demonstrates compatibility with both CNNs and Transformers, such as SwinIR.
>
> To address the reviewer’s concern on insufficient results of SwinIR, we provide the results of FitNet and FAKD for the X4 SwinIR model below for more comprehensive comparison. In the experiments of the feature-based KD methods, the feature maps after uniformly picked Swin Transformer layers are matched.
>
> | KD Method    | Set5             | Set14            | BSD100           | Urban100         |
> | --------- | ---------------- | ---------------- | ---------------- | ---------------- |
> | Logits-KD        | 32.27/0.8954     | 28.67/0.7833     | 27.62/0.7380     | 26.15/0.7887     |
> | FitNet    | 32.08/0.8925     | 28.51/0.7800     | 27.53/0.7354     | 25.80/0.7779     |
> | FAKD      | 32.06/0.8926     | 28.52/0.7800     | 27.53/0.7354     | 25.81/0.7780     |
> | **AugKD** | **32.41/0.8973** | **28.79/0.7860** | **27.69/0.7405** | **26.43/0.7972** |
>
> Additionally, we evaluated AugKD on a recent state-of-the-art SR network, Dense-residual-connected Transformer (DRCT) [1]. Using the large version of DRCT as the teacher model, we distill the DRCT network with different KD methods. The result again confirms that AugKD provides notable improvements for the student SR model:
>
> | Model (#Params.)         | Urban100     |
> | --------------------------- | ------------ |
> | Teacher: DRCT-L (27.58M) | 28.70/0.8508 |
> | Student: DRCT (10.44M)  | 27.23/0.8188 |
> | Logits-KD                   | 27.28/0.8195 |
> | FitNet                      | 27.21/0.8177 |
> | **AugKD**                       | **27.43/0.8226** |
>
> [1] Hsu, C. C., Lee, C. M., & Chou, Y. S. (2024). DRCT: Saving Image Super-resolution away from Information Bottleneck. arXiv:2404.00722.
>
> # Question 1: Clarification for the zoom-out operation in AugKD
>
> While the ground truth labels for  $I\_{\text{LR}\_{zo}}$ images are available, going through the teacher model in this path serves the critical purpose of effectively transferring teacher model's knowledge. In the KD of SR models, the teacher model's intended function, transferring knowledge to student, is shaded by the ground truth HR labels, due to the nature that the teacher model's outputs are noisy approximation to the high-resolution images, and there is barely "dark knowledge" like the inter-class relationship or label smoothing in the classification task. AugKD addresses this challenge by re-purposing the teacher model to provide supervision on auxiliary distillation samples that are efficiently generated and closely aligned with the original training data distribution. By matching the student and teacher models on both zoom-in and zoom-out auxiliary samples, AugKD enables the student model to extract meaningful priors encoded in the teacher’s outputs, and achieves more effective knowledge transfer.

---

### Meta-Review · Area_Chair_tJFa · 2024-12-22

**Metareview:**

This paper introduces AugKD, a novel knowledge distillation strategy for enhancing image super-resolution (SR) tasks. Its key technical contribution lies in the leverage of unpaired data augmentations to generate auxiliary distillation samples and enforce consistency regularization.

Overall, the reviewers appreciate the simplicity and effectiveness of the proposed method and commend the paper for being well-written and organized. But meanwhile, some concerns are raised, mainly regarding: 1) the focus of this paper (i.e., exclusively focusing on SR) may be somewhat narrow; 2) more recent advancements in SR should be compared; 3) how other data augmentations beyond zooming affect the performance; and 4) the motivations and insights of augmentations need to be further clarified.

These concerns are largely addressed in the rebuttal period, and all reviewers unanimously vote for acceptance. The AC supports this decision.

**Additional Comments On Reviewer Discussion:**

The major concerns are listed in my meta-review. During the discussion period, the authors provided additional experiments that address concerns (2) and (3). Regarding concerns (1) and (4), the authors offered further clarifications, such as clarifying that the proposed strategy could also benefit other low-level tasks, like denoising.

Overall, no major concerns remain after the discussion period, and all reviewers give this paper a positive rating.

---

### Decision · Program_Chairs · 2025-01-22

Accept (Poster)